# Molecular Insights into Inflorescence Meristem Specification for Yield Potential in Cereal Crops

**DOI:** 10.3390/ijms22073508

**Published:** 2021-03-29

**Authors:** Chengyu Wang, Xiujuan Yang, Gang Li

**Affiliations:** 1School of Life Sciences and Engineering, Southwest University of Science and Technology, Mianyang 621010, China; wcy863666492@163.com; 2School of Agriculture, Food and Wine, Waite Research Institute, Waite Campus, The University of Adelaide, Glen Osmond, SA 5064, Australia; xiujuan.yang@adelaide.edu.au

**Keywords:** inflorescence meristem, inflorescence architecture, shoot apical meristem, spikelet, branching, cereal crops, yield improvement

## Abstract

Flowering plants develop new organs throughout their life cycle. The vegetative shoot apical meristem (SAM) generates leaf whorls, branches and stems, whereas the reproductive SAM, called the inflorescence meristem (IM), forms florets arranged on a stem or an axis. In cereal crops, the inflorescence producing grains from fertilized florets makes the major yield contribution, which is determined by the numbers and structures of branches, spikelets and florets within the inflorescence. The developmental progression largely depends on the activity of IM. The proper regulations of IM size, specification and termination are outcomes of complex interactions between promoting and restricting factors/signals. Here, we focus on recent advances in molecular mechanisms underlying potential pathways of IM identification, maintenance and differentiation in cereal crops, including rice (*Oryza sativa*), maize (*Zea mays*), wheat (*Triticum aestivum*), and barley (*Hordeum vulgare*), highlighting the researches that have facilitated grain yield by, for example, modifying the number of inflorescence branches. Combinatorial functions of key regulators and crosstalk in IM determinacy and specification are summarized. This review delivers the knowledge to crop breeding applications aiming to the improvements in yield performance and productivity.

## 1. Introduction

Higher flowering plants display a huge diversity of inflorescence architectures ranging from a solitary flower to specialized structures that contain multiple branches and florets. This variation is temporally and spatially controlled by two opposite activities: the maintenance of meristems for new lateral organs and determinacy of meristems during the flower formation. The inflorescence meristem (IM), which derives from the shoot apical meristem (SAM), produces organs on its flanks while maintaining a pool of pluripotent stem cells at its apex [1,2]. The inflorescence of dicot plant *Arabidopsis thaliana* exhibits an indeterminate pattern, in which the IM constantly produces secondary inflorescence meristems, as well as floral meristems (FMs) [3]. In the grass family (*Poaceae*), inflorescence architecture is largely established by iterations of branching. The basal unit of grass inflorescence is referred to as a spikelet, which is a short and condensed branch containing leaf-like structures. Specifically, after the SAM converts into IM, the IM initiates branch meristems (BMs), and/or spikelet meristems (SMs), which further initiate FMs [4,5]. Modeling inflorescence meristems have been proposed as transitioning from an indeterminate IM to a determinate FM [4]. Consequently, the variation of this progression creates a wide range of inflorescence types exist among closely related species, such as panicle, raceme, spadix, and spike [4,6]. Some grass inflorescences, such as the rice panicle and maize tassel, consist of a main axis, long branches and spikelets, but others, like wheat and barley belonging to the Triticeae tribe, characteristically show an unbranched spike-type inflorescence that spikelets are directly attached to the inflorescence axis. The variation in inflorescence architecture is controlled by the activity of specialized meristems: the branch meristem and the spikelet meristem [6].

It is recognized that meristem determinacy is key to understanding inflorescence architecture [4]. The genetic basis of inflorescence initiation and development has been extensively studied in the eudicot *Arabidopsis* and monocot crops such as rice and maize [2,6]. For example, classic *Arabidopsis* CLV (CLAVATA)-WUS(WUSCHEL) negative regulatory loop controls SAM activity and size, and further affects IM formation and differentiation [1]; in rice and maize, regulatory mechanisms of conserved CLV signaling underlying meristem size control and inflorescence specification, have also been demonstrated [7]. Additionally, a large collection of grass genes and domestication QTL (quantitative trait loci) associated with inflorescence architecture has been identified [8,9,10,11,12,13,14,15]. The reported regulators are involved in peptide-receptor signaling, G protein pathway, plant hormone pathway, photoperiod signaling, transcription factor regulatory networks and microRNA-targets modules, which provide key insights into the molecular mechanisms regulating inflorescence architecture. Mutations of these players in grasses lead to the altered IM determinacy and size, resulting in the changed flower/spikelet number, inflorescence architecture, and final grain yield [1,3,5,6,7,16]. Importantly, these orthologous regulators and pathways among cereals show functional conservation and divergence in regulating inflorescence development and yield components. 

Enhancing the yield potential and stability of main cereals that are utilized as staple food and feed for humans and livestock, such as rice, maize, wheat, and barley, is a priority for global food security [17]. Crop selection and breeding have generated variants with increased yield through optimizing inflorescence traits, such as branching and spikelet number. Investigations of genes/alleles associated with inflorescence architecture have revealed basic developmental patterning mechanisms, which are pivotal for genetic approaches to optimize yield in cereal crops [12,13,14,15,18]. The role of spikelet development in yield improvement has been well-reviewed recently [19,20]. Genetic and genomic advances in cereal species have uncovered the molecular determinants of inflorescence architecture and facilitated knowledge transferring across genera. In this review, we focus on the developmental diversity of inflorescence meristem in cereals, firstly considering the regulation of the transition from SAM to IM. In the next section, we review the genes and regulatory pathways involved in differentiated fate of IM: transition to the BM or SM, and in determinacy of the SM. Then, we highlight the regulators involved in both the regulation of BM formation and SM specification. Additionally, we discuss how an understanding of the molecular mechanisms that control IM development could contribute to enhancing grain yield. Overall, this review aims to highlight the key players of inflorescence meristem regulation, and summarize the current knowledge, the molecular regulatory pathways and their crosstalk in cereal plants, providing the potential connections between meristem activity and grain yield.

## 2. The Structure and Developmental Fate of IM in Cereal Crops

The meristems of higher plants are centers of cell proliferation and organ initiation. Morphological studies have revealed that the SAM comprises a central zone of slowly dividing, undifferentiated cells and a peripheral zone of more rapidly dividing cells that are in transition towards specification [3]. In grass family, upon the transition from vegetative phase to reproductive phase, SAM ceases producing leaves on its flanks and turns into IM that, depending on the species, starts producing lateral branch meristems (BMs) or directly spikelet meristems (SMs). Therefore, neither the IM nor the BMs are directly converted to floret meristems (FMs). Instead, all the higher-order meristems produced by the IM and its branches are ultimately converted to SMs, which first produce two bracts known as glumes, followed by one or more FMs in each spikelet. Because the development of the spikelet is highly stereotyped and deterministic within most major groups of grasses, the spikelet is defined as the terminal differentiated unit of the inflorescence, rather than florets [4,6]. In other words, the IM produces either BMs or SMs on its flanks, and the BMs themselves further produce either secondary BMs or SMs. During this progression, different sets of developmental decisions lead to the diversity of inflorescence architecture in grasses [21,22]. 

Rice has a panicle-like inflorescence that gives rise to primary and secondary branch meristems (pBMs and sBMs) and further generates SMs and FMs, respectively, in a determinate pattern (Figure 1), producing an average of 150–250 grains per inflorescence in modern cultivars. More complicatedly, maize has two distinct inflorescences, tassel and ear, bearing male and female flowers, respectively. The apical tassel is derived from the SAM and consists of indeterminate two-rowed branches at its base and a many-rowed central spike, whereas an ear is positioned laterally in the axils of leaves and consists of a many-rowed axis bearing spikelets (Figure 1). Both tassel and ear have determinate paired SMs, each initiating two FMs [4,6,22]. One of the two pistillate florets in each spikelet of the ear is fertile and the other is aborted, and an ear is able to produce an average of 350–450 grains (Figure 1). In grass tribe Triticeae, like barley and wheat, inflorescences exhibit a branchless spike architecture with spikelets directly attached to the axis. In barley, the IM differentiates into many spikelet ridges and each one forms a final triple spikelet meristem (TSM) carrying three individual SMs [4,5]. The triple spikelet is composed of one central spikelet and two lateral spikelets, and the fertility of lateral spikelets is either suppressed to form a two-rowed inflorescence type with 20–35 grains per spike or promoted to form a six-rowed type carrying average 45–80 grains per spike (Figure 1). In wheat, the IM differentiates into axillary meristem (AM) that includes a lower leaf ridge and upper spikelet ridge. The spikelet ridge differentiates into an SM that forms several FMs in a distichous manner on the indeterminate rachilla [18]. Ultimately, each wheat spike produces average 50–80 kernels in most of modern cultivars. Importantly, distinct fates of meristems are decided by specific genetic control, molecular regulation and extrinsic environmental signals, leading to a great plasticity of grass inflorescences and yield variation.

## 3. The Genetic Regulation of Transition from SAM to IM

### 3.1. CLV Pathway

The SAM is in a balance between providing founder cells for new organs and maintaining its own stem cell niche. Manipulating the meristem size or breaking this balance could induce plants to produce more organs. The CLV-WUS negative feedback loop regulates the stem cell maintenance, which has been adequately investigated in *Arabidopsis*. The *WUS* gene, encoding a homeodomain transcription factor, induces expression of the *CLV3* gene. The CLV3 is a secreted peptide and is perceived by receptor complexes including leucine-rich repeats (LRRs) kinases CLV1 and CLV2 that, upon activation, stabilize the meristem stem cell population by signaling back to repress the expression of *WUS*, thereby completing the negative feedback loop [1,7]. Mutations of *CLV1*, *CLV2* and *CLV3* show increased IM size, resulting in increased numbers of flowers and floral organs [23,24,25].

The CLV signaling pathway is basically conserved in grass species (Figure 2 and Table 1) [7,26,27,28,29]. In rice, *FLORAL ORGAN NUMBER 1* (*FON1*) gene encodes a leucine-rich repeat receptor-like kinase similar to CLV1 in *Arabidopsis* [26]. *FON2* (also called *FON4*) is an ortholog of *Arabidopsis CLV3* [28,30]. The *fon2/4* mutants exhibit an increased size of SAMs and FMs, and then increased number of both primary branches and floral organs. By contrast, overexpression of *FON2* leads to a smaller IM with reduced floral organs, indicating that *FON2* has similar roles of CLV3 in rice. Genetic analysis of *fon1 fon2* double mutants suggests that *FON1* and *FON2* function in the same genetic pathway and FON2 may be the ligand of FON1 [30], indicating the conserved function of *CLVs* in limiting meristem size. Moreover, another two *CLV3*-like genes in rice, *FON2-LIKE CLEPROTEIN1* (*FCP1*) and *FCP2*, show similar expression patterns with broad accumulations in the SAM. Genetically, *FCP1* and *FCP2* function redundantly to maintain SAM activity independently from FON1 [31]. Notably, the interaction between *FON2* and the rice ortholog of *WUS*, named *TILLERS ABSENT1* (*TAB1*; also called *OsWUS*), happens in a slightly different scenario where *TAB1* is only required to maintain stem cells during axillary meristem development with *FON2* restricting its expression [32]. It is likely that the *FON2*-*TAB1* pathway has been recruited to play a specific role within a narrow developmental window in rice during evolution.

In maize, *THICK TASSEL DWARF1* (*TD1*) and *FASCIATED EAR2* (*FEA2*) genes are the closest orthologs of *Arabidopsis CLV1* and *CLV2*, respectively [27,33]. Mutations in these genes result in extensive overproliferation of stem cells and meristem fasciation, and the *td1 fea2* double mutant has more severe phenotypes than the single mutants [27]. In addition, *ZmCLE7* (CLAVATA3/ESR-related peptide) and *ZmCLE14*, have been identified as potential *CLV3* orthologs and both of them have a negative effect on SAM size [34].

Another maize *CLV3*-like gene, *ZmFCP1*, an ortholog to rice *FCP1*, also plays a key role in meristem specification, and *zmfcp1* mutants show a *clv*-like ‘fasciated ear’ phenotype. However, ZmFCP1 and ZmCLE7 signals are transmitted by FEA2 through different protein complex [35], suggesting that these CLV3-like peptides can parallelly regulate stem cell homeostasis in meristems. A new maize CLV receptor, FEA3, similar to FEA2 in sequence, has been reported to integrate into a mathematic model of the CLV–WUS feedback [34]. The *fea3* mutants show enlarged and fasciated meristems, and are partially insensitive to ZmFCP1 peptide treatment [34]. Although *fea2* mutants do not increase overall yield due to a compensatory reduction in kernel size, a decrease of *FEA2* expression level can increase IM size as well as kernel row number [36]. Recently, a weak allele of *FEA2* has been found having higher kernel row number and yield in maize [37]. *FEA3* acts in a separate pathway than *FEA2* but, similarly, the weak allele of *FEA3* increases kernel row number [34]. Therefore, the multiple-member CLV pathway has the potential to enhance grain yield in maize.

Although removing CLVs restrictions on meristems produces more spikelets, this effect is more likely due to increases of organs number rather than a prolonged indeterminate status of meristems, because higher order branches are rarely formed in maize ears of these mutants. Even though some *CLV*-like and *WUS*-like genes are highly expressed in developing barley inflorescence [38], the effects of the CLV pathway in wheat and barley inflorescence development are less reported. In wheat genome, 104 CLV3/CLE peptides have been identified. Phylogenetic analysis and chemically synthesized peptides treatment have revealed that these CLV3/CLEs may have distinct roles in regulation of root and shoot development [39]. All orthologs of CLVs and WUS in wheat and barley still remain unknown because of the lack of mutant resources. Future studies are required to explore CLV pathways and assess their contributions to grain yield in Triticeae crops.

### 3.2. KNOTTED 1-Like Homeobox (KNOX) Proteins

Class I KNOTTED 1-like homeobox (KNOX) proteins are key homeodomain transcription factors that regulate shoot apical meristem establishment and maintenance in plants [3,6]. In *Arabidopsis*, *SHOOT MERISTEMLESS* (*STM*) encodes a KNOTTED1 (KN1)-related homeodomain protein and *stm* mutants show very severe phenotype lacking the shoot meristem, which is fatal to seedlings [40]. Recent studies have uncovered that STM integrates into WUS-CLV loop to modulate the stem cell homeostasis [41]. In grasses, KNOX proteins are also required for the maintenance of the proper size and activity of the stem cell niche (Figure 2 and Table 1). Loss of function of *KN1* (*KNOTTED 1*) in maize leads to the defects of shoot meristem maintenance and inflorescence development [42]. Two maize BELL1-like homeobox (BLH) transcription factors, BLH12 and BLH14, act as cofactors of KN1 and accumulate in overlapping domains in shoot meristems. Similar to *kn1* mutants, *blh12 blh14* double mutants fail to maintain axillary meristems and develop abnormal tassels [43]. Consistently, mutation of *Oryza sativa HOMEOBOX 1* (*OSH1*) in rice, the maize *KN1* ortholog, shows abnormal SAM, smaller inflorescences and a decreased number of spikelets [44]. Rice *BELL1*-like homeobox genes are also involved in regulating inflorescence architecture and meristem maintenance [45]. Similarly, rice BELL1-type proteins form a heterodimer with KNOX-type proteins such as OSH1 [46]. It is likely that heterodimers of KNOX-type and BELL1-like proteins play a crucial role in maintaining the SAM in grasses. In wheat, the *KN1*-like homeobox gene, *WKNOX1*, is expressed in SAM-containing shoots and young spikes [47], but its genetic function is still unknown.

Exploring KN1 downstream targets is a potential way to understand the mechanisms of KNOX proteins in meristem maintenance. In maize, KN1 coordinates the regulatory gene networks by directly binding to a huge number of loci and genes, most of which encode key regulators involved in auxin, cytokinin, and gibberellic acid (GA) signaling pathways [48]. Maize KN1 represses the accumulation of bioactive GA directly through positive regulation of *GA2ox1* (*Gibberellin 2-beta-dioxygenase 1*) that encodes a GA-inactivating enzyme, which is required for proper establishment and maintenance of SAM and IM [49]. In rice, ectopic expression of KNOTTED1-like homeobox protein promotes the accumulation of cytokinin by activating cytokinin biosynthesis genes *adenosine phosphate isopentenyltransferases* (*IPTs*) [50]. Therefore, KNOX proteins may play a key role in regulating SAM activity by coordinating phytohormones in cereal plants (Figure 2), which is consistent with findings in dicot plants [51,52]. 

### 3.3. G-Protein Pathway 

Heterotrimeric G proteins contain Gα, Gβ and Gγ subunits and play a critical role in signal transmission [53]. Compared to the classic heterotrimeric G protein signaling in the mammalian system, plant G proteins are less understood [53,54]. Recently, several G protein subunits have been demonstrated contributing to meristem specification and inflorescence architecture in cereal plants (Figure 2 and Table 1). A maize Gα protein COMPACT PLANT2 (CT2) functionally interacts with the CLV receptor FEA2 to control SAM development [35,36]. In *ct2* loss-of-function mutants, SAM size is increased, and thicker tassels and denser ears are formed, which resembles *Arabidopsis clv* and maize *fea2* mutants, indicating that CT2 transmits a stem-cell-restrictive signal from a CLV receptor in maize [36]. Expression of a constitutively active CT2 results in the increased size of ear IMs, and higher spikelet density and kernel row number, all beneficial traits selected during maize improvement [55]. Knock out mutation of maize *Gβ* subunit (*ZmGB1*) shows a lethal phenotype, and genetic screening for suppressor of the lethal phenotype has revealed that ZmGB1 acts with Gα subunit gene to control meristem size and activity, indicating that Gβ and Gα are in a common signaling complex in maize. Moreover, ZmGB1 functions together with CT2 in downstream of the FEA2 CLAVATA receptor during inflorescence development (Figure 2) [56]. However, the role of maize Gγ subunit in shoot meristem development remains unknown. In rice, a Gγ subunit, rice *Dense and Erect Panicle 1* (*DEP1*), has been identified as a major QTL controlling panicle branching, seed size, and seed number, and wheat TaDEP1 also shows the similar function in regulating spike development [10,57]. Another rice G-protein γ subunit, Grain Size 3 (GS3), whose loss-of-function allele forms longer but fewer grains, acts the key regulator for grain shape and yield production. Supportively, natural variation in *GS3* can explain about 79% of the variation between short-grain versus long-grain cultivars [9]. Both Gγ subunits GS3 and DEP1 interact directly with a conserved floral homeotic E-class MADS-box protein, OsMADS1, to regulate grain size and shape [58]. In barley, a semi-dwarfing gene, *Brachytic1* (*Brh1*), encodes a G protein α subunit. *Brh1* mutation causes a shorter spike and rounded grain shape [59]. Except maize, currently there is a lack of proofs for heterotrimeric G protein subunits from other cereal crops participating in the crosstalk with CLV pathways.

### 3.4. Genetically Controlled Photoperiod Response in Meristem Specification

The transition from SAM to IM in cereals is also controlled by orchestration and integration of endogenous signals and exogenous signals, such as temperature and photoperiod [5,6,18]. In molecular level, this transition is accomplished by the florigen activation complex (FAC) (Figure 2). For example, rice florigen FT (FLOWERING LOCUS T)-like protein Hd3a binds its cofactor 14-3-3 protein to form a complex which further recruits bZIP transcription factors, like OsFD1 (FLOWERING LOCUS D 1) and OsFD4, to promote the phase transition [60,61]. The photoperiod-dependent accumulation of florigen protein at apical meristem is critical for the activity of FAC. Hence, upstream regulators of florigen are directly or indirectly affecting apical meristem activity. Heading date 1 (Hd1), an upstream regulator of Hd3a, the rice ortholog of *Arabidopsis* photoperiod response regulator CONSTANS (CO), is an enhancer of phase transition under short-day conditions [8]. Rice Early heading date 1 (Ehd1) is a type-B response regulator and promotes floral transition under short-day conditions through upregulating *Hd3a* expression [62,63]. Grain number, plant height, and heading date 7 (Ghd7) acts as a floral repressor inhibiting Ehd1 in the context of phytochrome signals [64]. Monocot-specific zinc-finger transcription factors, Ehd2 of rice and INDETERMINATE 1 (ID1) of maize, also have been reported to regulate meristem activity by controlling photoperiodic pathway through Ehd1 [65,66]. Importantly, impacts of these photoperiodic regulators appear beyond the phase transition at apical meristem. Enhanced expression of *Ghd7* increases the number of secondary branches and panicle size in rice under long-day conditions [67]. Overexpression of the maize photoperiod response gene, *ZmCCT10* (*CO*, *CONSTANS*, *CO-LIKE and TIMING OF CAB1*), an ortholog of the rice *Ghd7*, modifies flowering time and inflorescence morphology [68]. Naturally occurring rice genetic alleles of *Hd1*, *Ehd1*, *Ghd7* and *Ghd8*, have been used in breeding aiming to higher grain yield [11,63,67,69], yet how these photoperiodic genes regulate inflorescence meristem specification afterwards remains largely unknown.

In Triticeae, functionally homologous regulators of flowering have also been investigated (Figure 2). Wheat *VRN3* (*VERNALIZATION 3*) gene encodes an ortholog of FT protein, and mutations of this gene in bread and tetraploid wheat delay the transition to reproductive growth, prolong the duration of spike development, and increases the number of spikelets [70,71]. Overexpression of a barley *FT*-like gene (*HvFT4*) specifically delayed spikelet initiation and reduced the number of spikelet primordia and grains per spike [72]. Significantly, both wheat and barley have a major QTL contributing to photoperiodic regulation of flowering called *Photoperiod*-1 (*Ppd-1*) [73,74]. Wheat Ppd-1 influences inflorescence architecture and paired spikelet development by modulating the expression of *FT1* under short day condition [74]. Moreover, Ppd-1 activates *FT2*, and the latter regulates the number and formation of spikelets [75]. PHYTOCHROME C (PHYC) acts upstream activating *Ppd-1* and *FT1* in tetraploid wheat, and the *phyc* mutant delays flowering and alters spike development [76]. Additionally, wheat Teosinte Branched1 (TaTB1), a TCP (Teosinte branched1/Cincinnata/Proliferating cell factor) family transcription factor, is decoupled from photoperiod but can interact with the florigen protein, FT1, and regulate inflorescence branching [77]. In barley, a gene network closely associated to Ppd-1 has been investigated, including *FT2*, floral homeotic MADS-box genes like *SEPALLATA1* (*SEP1*), *SEP3*, *PISTILLATA* (*PI*), *APETALA3* (*AP3*), and *VRN1* [78]. These genes are highly expressed in developing IMs [38]. Ppd-1 is also involved in the crosstalk with vernalization flowering pathway. Together with CO homologs in barley, HvCO1 and HvCO2, Ppd-1 represses flowering by up-regulating *HvVRN2* (the ortholog of rice *Ghd7* and wheat *TaVRN2*) expression before vernalization but promotes *HvFT1* after vernalization [79,80]. During the phase transition of the apical meristem, major cereals share common regulatory patterns in response photoperiod. More importantly, the flowering regulators often continue playing an active role in the IM development, therefore contributing to yield production.

### 3.5. Other Pathways

As an outstanding family of regulators active throughout the entire reproductive development, MADS-box transcription factors are closely associated with the phase transition of grass inflorescences [6]. For instance, knockdown of three *APETALA 1* (*AP1*)/*FRUITFULL* (*FUL*) family members, *OsMADS14*, *OsMADS15*, and *OsMADS18*, in a rice *sepallata* (*SEP*) mutant *panicle phytomer 2* (*pap2*) background shows a delayed transition from SAM to IM and produce multiple shoots instead of one inflorescence (Table 1) [81]. As to Triticeae, *VRN1*, encoding a MADS box transcription factor, is induced by vernalization to trigger meristem transition and flowering in wheat and barley [82]. Ectopic expression of barley VRN1 protein accelerates the transition to reproduction and flowering [83]. Wheat TaVRN1 cooperates with another SVP (SHORT VEGETATIVE PHASE) -like MADS protein to regulate vernalization-induced reproductive transition [84]. In addition, wheat MADS-box genes, *VRN1*, *FUL2* and *FUL3* have redundant roles in regulation of the transitions from the vegetative SAM to IM and from IM to spikelet. The *ful2* null mutant produces more florets per spikelet, additive to a higher number of spikelets, resulting in a significant increase in the number of grains per spike in the field [85]. Moreover, complexes of wheat FUL and SVP act during the early reproductive phase to promote heading and formation of the spikelet [86]. Thus, these *FUL* family genes are essential in the acquisition and termination of IM identity (Figure 2). Whether other MADS-box genes also contribute to the transition from SAM to IM in cereal crops needs further investigation.

Besides as components of FAC, bZIP transcription factors also regulate meristem activities via other pathways. Loss of function of maize *FEA4* leads the increased meristem size and similar fascitated phenotype from maize *fea2* and *fea3* mutants [87]. FEA4 promotes expression of many genes involved in meristem determinacy and auxin signaling, which shares some targets with KN1. Therefore, FEA4 and KN1 may act antagonistically, in controlling the determinacy–differentiation balance [87]. FEA4 activity is controlled by a redox mechanism, via interacting with the glutaredoxin MALE STERILE CONVERTED ANTHER1 (MSCA1). Dominant mutations in *MSCA1* show bigger meristems and loss-of-function mutants of *msca1* correspondingly have smaller shoot meristems [88]. Redox signaling also play a key role in the controlling of shoot meristem size in other plants by regulating *WUS* expression [89]. 

Plant hormone cytokinin is essential in the regulation of meristematic activity, inflorescence branching in plants [1]. Mutations in rice *OsCKX2* (a cytokinin oxidase/dehydrogenase) and *LONELY GUY* (*LOG*, a cytokinin-activation enzyme) could lead to altered cytokinin distribution in meristems and consequently change the SAM, IM and BM activities [90,91]. In barley, the dynamic of cytokinin is also required for inflorescence meristem maintenance and spike architecture [92]. Modification of cytokinin content via manipulating the key cytokinin oxidase/dehydrogenase genes has become a potential way for yield improvement in wheat and barley [93,94].

## 4. IM Differentiation: Branches or Spikelets 

After the initiation of IM, patterns of determinacy in IM shape the inflorescence morphology. The developmental fate of the IM in grasses, i.e., its conversion into a SM or its production of branch meristems that convert to SMs later, is species specific and determines the architecture of inflorescence [4]. In rice, before completing the terminal SM development, the whole IM maintains an indeterminate status, generating primary and secondary branches, and forms a branching architecture (panicle) [6]. While meristem determinacy in wheat and barley is directly caused by the SM identity, which ultimately produces florets without branches. Wheat inflorescence is determinate and produces a terminal spikelet at the apex, whereas the barley inflorescence is indeterminate [5,18]. Therefore, IM determinacy and maintenance control the numbers of branches and spikelets. Multiple factors and pathways synergistically regulate the IM specification and activity (Figure 3 and Table 2), further affecting yield performance in cereal crops. 

### 4.1. The MADS—RICE CENTRORADIALIS (RCN) Pathway Mediated IM Differentiation on Inflorescence Branching 

MADS-box transcription factors play essential roles in floral organ determination and inflorescence architecture in plants. In rice, *PAP2*/*OsMADS34*, encoding a SEPALLATA-like MADS-box transcription factor, is a core component of the floral identity complexes, similar to its orthologs in dicots [96,97]. Mutation of rice *PAP2* causes more primary branching events and disorganization of spikelet morphology [81,98], suggesting *PAP2* inhibits overgrowth of new organs from IM. Consistently, overexpressing wheat *TaPAP2-5A*, an ortholog of rice *PAP2,* inhibits SM formation and leads to the reduced numbers of spikelets per spike [99]. Importantly, the function of *SEPALLATA* genes in inflorescence branching also has been revealed in other crops, such as tomato (*Solanum lycopersicum*) and foxtail millet (*Setaria italica*) [100,101], suggesting the conserved role of *SEPALLATA* genes in plant IM identity.

Rice *RCN1* (RICE CENTRORADIALIS1) and *RCN2* are orthologs of the *Arabidopsis TERMINAL FLOWER1* (*TFL1*) gene, a key regulator of inflorescence architecture [102]. Overexpression of *RCN1* and *RCN2* leads to enlarged IM and more secondary branches, which alter the panicle morphology in rice [103]. RCNs maintaining the indeterminacy of IM is likely via antagonizing FT-like protein Hd3a for competitive binding to 14-3-3 and OsFD1 [95]. Similarly, ectopic expression of the *TFL1*-like genes, *ZCNs* (*ZEA CENTRORADIALIS*) increases branching and spikelet number of tassels in maize [104]. Wheat *TaTFL1* is an ortholog of *TFL1/RCN*, and overexpression of *TaTFL1-2D* in wheat causes the increased numbers of spikelets, indicating that TaTFL1 also regulates spike complexity in wheat [99]. Additionally, it has been demonstrated that SEPALLATA/PAP2 can bind to the promoter of *TFL1*/*RCN* and directly suppress its expression in both *Arabidopsis* and rice (Figure 3) [105]. The similar regulatory module has also been proposed in *Setaria italica* [101]. Thus, the MADS-RCN pathway in regulating the meristem identity and inflorescence architecture is likely conserved in grass family.

### 4.2. The Role of RAMOSA (RA) Genes in Inflorescence Branching

Inflorescence branch number in cereal crops has long been recognized as an important trait on grain yield. In maize, inflorescence architecture is determined by the collective actions of *RA* genes (Figure 3). The *ramosa 1-3* (*ra1-3*) mutants produce a highly branched inflorescence, where spikelet pairs are replaced by long branches, suggesting that these genes function in regulating branch number. The various degrees of increased branching observed in *ramosa* mutants indicate potential regulatory pathways composed of *RA* genes on meristem determinacy [106,107,108]. Moreover, mutation of *RA2* or *RA3* leads to the reduced *RA1* expression indicating that they act upstream of *RA1* and promote its expression. *RA1* encodes a Cys2-His2 zinc finger transcription factor, *RA2* encodes a LATERAL ORGAN BOUNDARY (LOB) domain transcription factor, and *RA3* encodes a trehalose-6-phosphate phosphatase (TPP) metabolic enzyme [106,107,108]. Coding sequence alignment has shown that *RA2* is conserved in grasses, but *RA1* is not [109]. In rice, *OsRA2*, the ortholog of maize *RA2* gene, modifies panicle architecture through regulating pedicel length, while an ortholog of maize *RA1* does not exist in rice genome [109]. Maize RA3 regulates inflorescence branch by manipulating the sugar signal that moves into axillary meristem [4,108]. Another maize trehalose-6-phosphate phosphatase, TPP4, is also associated with inflorescence development, and loss of *TPP4* leads to reduced meristem determinacy and more inflorescence branching. The *ra3 tpp4* double mutants have demonstrated that TPP4 acts as a redundant backup for RA3 [110]. Surprisingly, analysis of an allelic series has been revealed that TPP enzymatic activity is decoupled from its signaling transmission, as a catalytically inactive RA3 can complement *ra3* mutant suggests that a non-enzymatic function is responsible for its role in meristem determinacy [110].

Barley Six-rowed spike 4 (VRS4), an ortholog of the maize RA2, controls barley inflorescence row type and spikelet meristem determinacy (Figure 3 and Table 2) [5,111]. In *vrs4* mutants, SMs frequently produce branch-like meristems, developing a spike-branching architecture. Additionally, VRS4 regulates the expression of *SISTER OF RAMOSA3* (*HvSRA3*) and a trehalose biosynthetic enzyme trehalose-6-phosphate synthase gene (*HvTPS1*), suggesting the crosstalk between RA2 and RA3 in inflorescence branching [111]. Barley VRS4 also regulates the *Hordeum* specific row-type pathway through VRS1, a homeodomain-leucine zipper transcription factor [111]. Thus, RA2 may have a partially conserved function in repressing inflorescence branch outgrowth in grasses. The comparison of RAMOSA pathways between barley and maize indicates an ancestral and long evolutionary history of inflorescence branching, but a recent diverged evolutionary event of row type, which appears to improve the understanding of profound structural differences in cereal inflorescences.

### 4.3. Conserved Function of FRIZZY PANICLE (FZP) in SM Identity

*FZP*, encoding an AP2/ERF transcription factor, is a positive regulator of the transformation from BM to SM (Figure 3). In rice, *FZP* restricts axillary meristems formation and promotes floral meristem identity and maintenance [112]. The ortholog of *FZP* in maize, is called *BRANCHED SILKLESS1* (*BD1*). The *bd1* mutants produce more than two glumes, and these glumes fail to inhibit axillary meristems, resulting in new spikelets generated from the axils [113]. Moreover, *FZP* orthologs in Triticeae, barley *COM2* (*COMPOSITUM 2*) and wheat *BRANCHED HEAD1* (*TtBH1*) and *WFZP* (*wheat FZP*) [114,115,116], have also been reported to regulate spike architecture, suggesting a conserved function of *FZP* in cereal species. Missense mutations in the *TtBH1* and *COM2* create ‘Miracle wheat’ and ‘Compositum-barley’ plants with highly branched inflorescences, respectively. These mutations facilitate the development of inflorescence like structures at rachis nodes where a spikelet typically forms, suggesting that the functions of *TtBH1* and *COM2* are required for the axillary meristems to acquire the spikelet identity [116]. In addition, barley *COM2* may act downstream of VRS4 [116]. Another *FZP* ortholog in wheat, *WFZP*, is specifically expressed at the sites where the SM and FM are initiated, which differs from the expression patterns of its orthologs *FZP/BD1* in rice and maize, implying its functional divergence during species differentiation. WFZP directly promotes the expression of *VRN1* and *HOMEOBOX4* (*TaHOX4*) to inhibit axillary meristem generation and promote SM-to-FM transition [117].

FZP also shows a precise mechanism for grain production by modifying SM formation in cereals. Among mutants of *FZP* orthologs, a greater grain yield is found in tetraploid ‘Miracle Wheat’ and hexaploid wheat but not in diploid barley, rice and maize. In hexaploid wheat, combinations of a *wfzp-d* null mutation with a frame-shift mutation in *wfzp-a* can produce a more severe supernumerary spikelet phenotype, in comparison with the wild type *WFZP-A* [114]. Despite a lower level of *WFZP-A* than *WFZP-D* in hexaploid wheat, such a buffering effect of polyploidy on mutations suggests that low levels of *FZP* or its orthologs are sufficient to increase numbers of both inflorescence branches and spikelets. The favorable alleles of *WFZP* associated with spikelet number per spike (SNS) are preferentially selected during breeding [117]. Rice FZP plays an important role in the transition from fewer secondary branches in wild rice to more secondary branches in domesticated rice cultivars, and variation in the regulatory promoter region of *FZP* causes its decreased expression level and increases the secondary inflorescence branching and grain yield [118], suggesting that the delicate expression level of *FZP* is required for agricultural performance. Thus, selection of variations in the *cis*-regulatory region of *FZP* may open a new access to crop domestication and improvement. Fine-tuning *FZP* expression level by beneficial alleles may be an efficient approach to further improve yield in cereals.

### 4.4. Regulation of TCP Transcription Factors in Inflorescence Architecture

Plant-specific transcription factors TCPs that share a conserved bHLH motif, are key regulators of lateral organs, including inflorescence branching and tillering [1]. *TEOSINTE BRANCH1* (*TB1*) encodes a CYC (CYCLOIDEA) type TCP protein, a gene first cloned in maize, regulating tillering and ear size [119]. This gene is one of the key genetic loci for maize domestication, and TB1 directly or indirectly regulates several MADS-box genes in controlling both plant and inflorescence architecture [120]. Maize *ZmBAD1* (*BRANCH ANGLE DEFECTIVE 1*) also encodes a TCP protein that promotes cell proliferation in the pulvinus and influences inflorescence architecture by impacting the angle of lateral branch emergence [121]. In other cereal crops, the role of TB1 orthologs in repressing axillary bud outgrowth is also well studied [1,20]. Rice OsTB1 negatively regulates lateral branching, including both tillering and branching of the panicle [122]. Another *TB1* homolog in rice, *OsTB2/RETARDED PALEA1* (*REP1*), has been reported affecting palea development, recently been found associated with tiller number during rice domestication [123,124]. The wheat *TB1* ortholog, *TaTB1*, regulates both tillering and branching of spikes in bread wheat, and the increased dosage of *TaTB1* promotes inflorescence branching and delays inflorescence growth by reducing the expression of meristem identity genes at early developmental stages [77]. Barley *INT-C* (*INTERMEDIUM-C*, also known as *VRS5*), an ortholog of *TB1*, suppresses development and outgrowth of the lateral spikelets, but not inflorescence branching [125]. Shang et al. (2020) and Poursarebani et al. (2020) have reported a CYC/TB1-type TCP transcription factor, *BRANCHED AND INDETERMINATE SPIKELET 1* (*BDI1*)/*COMPOSITUM 1* (*COM1*), in controlling inflorescence branching via specifying meristem identity. *bdi1/**com1* mutants produce branch-like structures instead of floret, indicating that BDI1/COM1 confers SM identity [126,127]. The expression of genes involved in cell wall development, hormone signaling, and carbohydrate-based metabolic processes are changed in *bdi1/**com1* mutants, which is consistent with the phenotypes of spikelet meristem determinacy and cell boundary formation [126,127]. One potential target gene of VRS4, *HvSAR3*, is also down-regulated in *bdi*/*com1* mutants, suggesting the crosstalk between TB1 and RAMOSA pathways [108,111,126,127,128]. Interestingly, the closest orthologs of BDI1/COM1 are rice OsTB2/REP1 and maize ZmBAD1, OsTB2/REP1 is involved in controlling palea development and tiller number, but not inflorescence branching [123,124], while barley BDI1 and maize ZmBAD1 have no reported effect on palea development or tiller outgrowth [121,127,128]. Thus, TCP family is essential in regulation of lateral tissue development and outgrowth of vegetative tillers, reproductive organs and inflorescences, but orthologs show slightly diverse functions in inflorescence and floret development among cereals.

### 4.5. Other Key Regulators/Modules Involved in BM and SM Identify

*ABERRANT PANICLE ORGANIZATION 1* (*APO1*), and *APO2*, the orthologs of *Arabidopsis UNUSUAL FLORAL ORGANS* (*UFO*) and *LEAFY* (*LFY*), act as key inflorescence architecture regulators, suppressing specification of SM in rice [129,130]. APO2 physically interacts with APO1, and the function of APO1 depends on APO2, as shown by genetic and biochemical assays [130]. Recently, it has been reported that APO1 and APO2 physically associate with an E3 ubiquitin ligase, LARGE2, to regulate panicle size and grain number, suggesting a promising module of LARGE2-APO1/APO2 for yield improvement in crops (Figure 3) [131]. Wheat *TaAPO-A1*, an ortholog of rice *APO1*, is crucial for the total spikelet number determination, suggesting the conserved nature of this gene across grass species [132]. Rice *TAW1* (*TAWAWA1*) encodes a nuclear protein and acts the upstream of *SVP* MADS-box genes to fine-tune the timing when meristems acquire the determinant spikelet phase. A dominant gain-of-function mutation of *TAW1* exhibits prolonged IM activity and branch formation, and delayed spikelet specification, hence increased spikelet number, in contrast, reductions in *TAW1* expression cause precocious IM abortion, resulting in the generation of small panicles with reduced primary branches [133]. 

*IDEAL PLANT ARCHITECTURE 1* (*IPA*1), encoding a rice Squamosa Promoter Binding Like Protein 14 (OsSPL14) that is targeted by miR156 (microRNA156), controls the spikelet transition by promoting the conversion from BM to SM [134,135]. Manipulating *IPA1* expression to an optimal dose leads to ideal yield, demonstrating a practical approach to efficiently design elite super rice varieties [136]. Another two targets of miR156, *OsSPL4* and *OsSPL17*, also positively regulate the transition from branch to spikelet meristem in rice [135]. Similarly, expressing wheat *TaSPL13* leads to increased panicle branches in rice, and increased florets and grains per spike in wheat [137]. Additionally, miR172 represses *AP2* (*APETALA2*)-like transcription factors through transcript cleavage and translational repression, which functions in controlling inflorescence and spikelet development in cereals (Figure 3 and Table 2) [135,138,139]. In rice, *miR172* and several *AP2* genes, including *OsIDS1* (*INDETERMINATE SPIKELET 1*) and *SUPERNUMERARY BRACT* (*SNB*)*,* control panicle branching and spikelet meristem identity [135,140]. Mutations of *AP2* genes result in reduced number of branches and compromised spikelet structure with extra bract-like organs [135,140]. Maize *AP2* genes, *IDS1* and the *Sister of IDS1* (*SID1*), play similar roles with their counterparts of rice in controlling branching and SM determinacy [138,139]. A key domestication-related gene in wheat, *Q* (*AP2L5*), an ortholog of maize *IDS1*, promotes spikelet numbers in domesticated spikes [16,141,142]. Null mutations of *Q* lead to reduced spikelet number per spike (SNS). Meanwhile, elevated levels of *Q* and mutations within its miR172-binding site increase SNS [143,144]. Beyond spikelet determinacy, miR172-targeted *Q* and its paralog *AP2L2* further redundantly determine floret identity by preventing the retrogression from florets to glumes [143,144]. In barley, suppression of miR172 guided cleavage of *AP2* mRNA also affects spikelet determinacy and spike architecture [145,146]. Barley *INT-M/DUB1* (*INTERMEDIUM-M/DOUBLE SEED1*), encoding an AP2L transcription factor (HvAP2L-H5) orthologous to maize IDS1 and wheat Q, is required for barley spike indeterminacy and spikelet determinacy via promoting IM activity and the maintaining meristem identity. Mutations in *INT-M/DUB1* lead to the decreased SNS but extra florets per spikelet [147]. Thus, the functions of *AP2*-like genes in spikelet identity and meristem activity are likely conserved in cereal crops, and the module of miR172-AP2 is crucial for the correct development of spikelets and florets. Moreover, SPLs modulate inflorescence development by regulating the miR172/AP2 pathway in rice and barley, and PAP2-RCN1 module in rice (Figure 3) [135,148], suggesting the regulatory networks of microRNAs during cereal inflorescence development. The manipulation of these regulatory modules may provide an opportunity to modify inflorescence architecture and improve grain yield in cereals.

## 5. Conclusions and Future Perspectives

In conclusion, genetic findings have identified multiple genes and regulatory pathways that control inflorescence meristem specification and fate (Table 1 and Table 2), which underpin key domestication-related traits of inflorescence architecture in cereals. Comparisons of the well-identified regulatory components of CLV signaling, G-protein signaling, plant hormone pathway, photoperiodic signaling, the RAMOSA pathway and other regulatory modules in rice, maize, wheat and barley, have significantly deepened to our understanding of inflorescence development at molecular level (Figure 2 and Figure 3). Due to the similar progression of spikelet determination among cereals, it is a powerful tool to exploit the homology of genes and pathways that are involved in spikelet development/number control. At the same time, some significant differences in inflorescence architecture also exist in cereal crops, such as the highly branched panicle in rice versus unbranched spike in barley and wheat, pointing to the distinct branch meristem identity controlled by several key transcription factors, like FZP and TCP. It will be exciting to further explore the mechanisms underlying the diversity of inflorescence architecture and meristem activity among cereals.

Given that multiple regulators, such as *IPA1*, *DEP1* and *FZP* in rice, *TB1*, *FEA2*, *FEA3* in maize, *Q* locus in wheat, *VRS5* in barley, have been selected for high-yield breeding during domestication, a rational design to create defined ideotypes using cutting-edge technologies has been proposed as future breeding strategies [149]. A large number of QTL that control inflorescence architecture traits have been identified and cloned in rice and maize. However, direct cloning of yield- and architecture-related genes using strategies of gene high-resolution mapping and map-based cloning in wheat and barley is still very difficult due to the complexity of their genomes. Importantly, the availability of the sequenced genome and pan-genome resources creates new possibilities of gene identification and gene function analysis [150,151,152,153]. Further researches that take advantage of novel technologies, such as CRISPR/Cas9 and single-cell RNA sequencing [154,155], are promising to provide valuable insights into the genes that control meristem identity and inflorescence architecture in cereals. The accumulating knowledge on inflorescence development can be harnessed to boost the yield potential of crops.

## Figures and Tables

**Figure 1 ijms-22-03508-f001:**
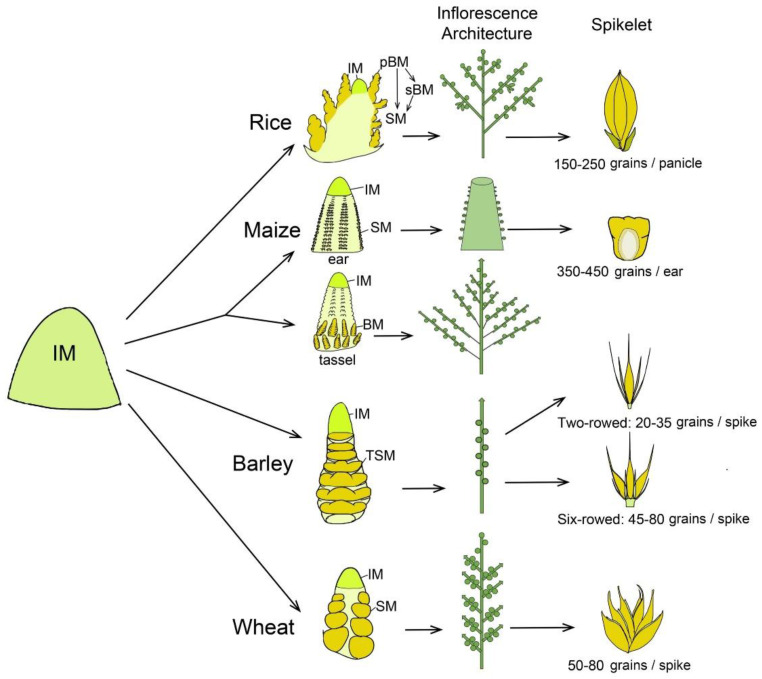
Schematic representation to compare inflorescence meristem differentiated fate, inflorescence architecture and spikelet in rice, maize, barley and wheat. In rice, inflorescence meristem (IM) generates two types of lateral branch meristems (BMs). The primary branch meristems (pBMs) generate spikelet meristems (SMs) and secondary branch meristems (sBMs). The sBMs further produce more SMs. In maize, IM of tassel is converted from shoot apical meristem, while the axillary meristem converts into an ear. The ear IM initiates a series of determinate axillary meristems, giving rise to pairs of SMs. The tassel produces BMs, which then form pairs of SMs. Each SM of ear and tassel further initiates two floret meristems (FMs). In barley and wheat, the IM directly differentiates SMs in axis without forming BMs. Barley has a triple spikelet meristem (TSM) structure composed of a central spikelet and two lateral spikelets, whose development is either suppressed to form a two-rowed type or promoted to form a six-rowed type. Conversely, in wheat, the inflorescence is composed of single spikelet that produce multiple FMs.

**Figure 2 ijms-22-03508-f002:**
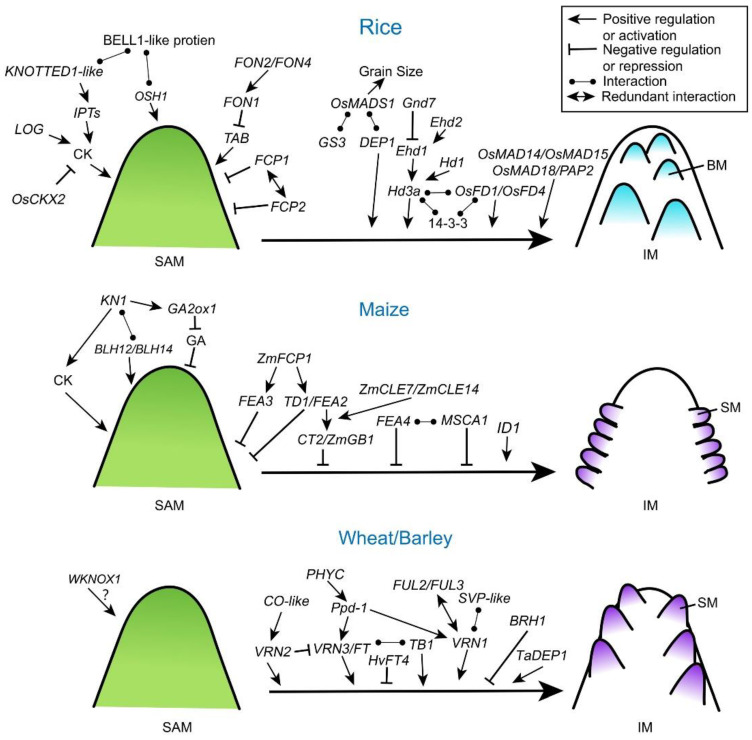
Key regulators and genetic pathways of phase transition from SAM to IM in cereal crops. Models for the roles of KNOX-type proteins, CLV signaling, G proteins, photoperiod pathway and MADS transcription factors in regulation of SAM size/activity and IM specification in rice, maize, wheat and barley. SAM, shoot apical meristem; IM, inflorescence meristem; BM, branch meristem; SM, spikelet meristem.

**Figure 3 ijms-22-03508-f003:**
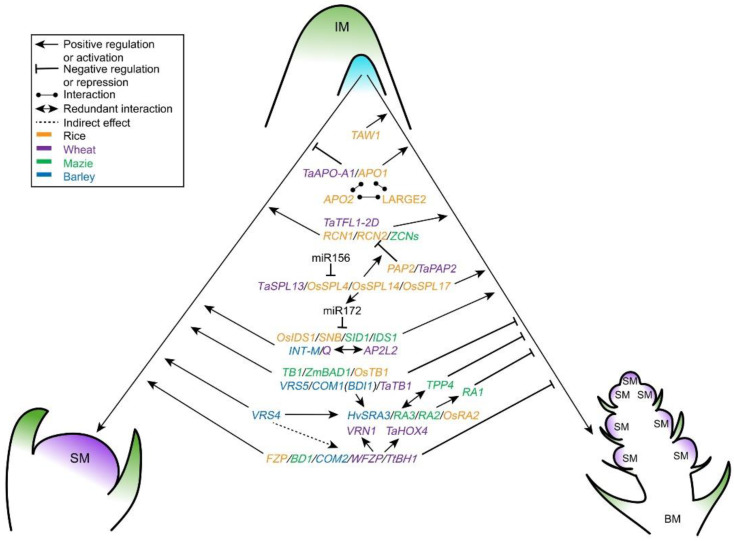
Genetic regulation of IM differentiation in cereal crops. The multiple players, including MADS, TCP, AP2 and SPL transcription factors, RA proteins, miRNAs, and their crosstalk genetically control BMs or SM identity in rice, maize, wheat and barley. IM, inflorescence meristem; BM, branch meristem; SM, spikelet meristem.

**Table 1 ijms-22-03508-t001:** Key regulators involved in the transition from SAM to IM in rice, maize, barley and wheat.

Rice	Maize	Barley	Wheat	Pathways	Reference
*FON2/4*	*ZmCLE7*; *ZmCLE14*			CLV-WUS	[28,30,34]
*FCP1*; *FCP2*	*ZmFCP1*			CLV-WUS	[31,34]
*FON1*	*TD1*			CLV-WUS	[26,27]
	*FEA2*			CLV-WUS	[33]
	*FEA3*			CLV-WUS	[34]
*TAB1*				CLV-WUS	[32]
*OSH1*	*KN1*		*WKNOX1*	KNOX	[42,44,47]
	*BLH12*; *BLH14*			KNOX	[43]
	*CT2*			G-protein	[35,59]
	*ZmGB1*			G-protein	[56]
*GS3*			*TaDEP1*	G-protein	[9,10,57]
	*Maize Gα*	*Brh1*		G-protein	[55,59]
*Hd3a*		*HvFT1*; *HvTF2*; *HvFT4*	*VRN3*; *FT1*; *FT2*	Photoperiod	[60,70,71,72,74,75,78]
*Ehd1*				Photoperiod	[62,63]
*Hd1*		*HvCO1*; *HvCO2*		Photoperiod	[8,80]
*Ghd7*	*ZmCCT10*	*HvVRN2*	*TaVRN2*	Photoperiod	[67,68,79,80]
*OsFD1; OsFD4*				Photoperiod	[61,95]
*Ehd2*	*ID1*			Photoperiod	[65,66]
		*Ppd-H1*	*Ppd-1*	Photoperiod	[73,74]
*OsMADS1*; *OsMADS14*; *OsMADS15*; *OsMADS18*		*HvVRN1*	*FUL2*; *FUL3*; *TaVRN1*	Others	[81,83,84,85]
	*FEA4*			Others	[87]
	*MSCA1*			Others	[88]
*OsCKX2*; *LOG*		*HvCKXs*	*TaCKXs*	Others	[90,91,93,94]

Abbreviations of gene names: *FON*, *FLORAL ORGAN NUMBER*; *CLE*, *CLAVATA3/ESR-related*; *FCP*, *FON2-LIKE CLE PROTEIN*; *TD1*, *THICK TASSEL DWARF1*; *FEA*, *FASCIATED EAR*; *TAB1*, *TILLERS ABSENT1*; *OSH1*, *Oryza sativa HOMEOBOX 1*; *KN1*, *KNOTTED1*; *BLH*, *BELL1-like homeobox protein*; *CT2*, *COMPACT PLANT2*; *ZmGB1*, *Zea mays Gβ subnuit*; *GS3*, *Grain Size 3*; *DEP1*, *DENSE ERECT PANICLE1*; *Brh1*, *Brachytic1*; *Hd3a*, *Heading date 3a*; *FT*, *FLOWERING LOCUS T*; *VRN*, *VERNALIZATION*; *Ehd1*, *Early heading date 1*; *Hd1*, *Heading date 1*; *CO*, *CONSTANS*; *ZmCCT10*, *CO*, *CONSTANS*, *CO-LIKE and TIMING OF CAB1*; *Ghd7*, *Grain number*, *plant height*, *and heading date 7*; *FD*, *FLOWERING LOCUS D*; *ID1*, *INDETERMINATE 1*; *Ppd*, *Photoperiod-H1*; *FUL*, *FRUITFULL*; *MSCA1*, *MALE STERILE CONVERTED ANTHER1*; *CKX*, *Cytokinin Oxidase/dehydrogenase*; *LOG*, *LONELY GUY.*

**Table 2 ijms-22-03508-t002:** Key regulators involved in IM differentiation and specification in rice, maize, barley and wheat.

Rice	Maize	Barley	Wheat	Pathways	Reference
*PAP2/OsMADS34*			*TaPAP2*	MADS-RCN	[96,97,99]
*RCN1*; *RCN2*	*ZCNs*		*TaTFL1*	MADS-RCN	[99,103,104]
	*RA1*			RAMOSA	[106]
*OsRA2*	*RA2*	*VRS4*		RAMOSA	[107,109,111]
	*RA3*	*SRA3*		RAMOSA	[108,111]
	*TPP4*			RAMOSA	[110]
*FZP*	*BD1*	*COM2*	*TtBH1*; *WFZP*	FZP	[112,113,115,116]
*OsTB1*; *OsTB2/REP1*	*TB1*; *ZmBAD1*	*VRS5*; *COM1/BDI1*	*TaTB1*	TCP	[77,119,121,122,123,125,126,127]
*APO1*; *APO2*			*TaAPO-A1*	Others	[129,130,132]
*TAW1*				Others	[133]
*OsSPL14*; *OsSPL4*; *OsSPL17*			*TaSPL13*	Others	[134,135,137]
*SNB*; *OsIDS1*	*IDS1*; *SID1*	*INT-M/DUB1*	*AP2L2*; *Q*	Others	[138,139,140,143,144,147]

Abbreviations of gene names: *PAP*, *PANICLE PHYTOMERL*; *RCN*, *MADS—RICE CENTRORADIALIS*; *ZCN*, *ZEA CENTRORADIALIS*; *TFL1*, *TERMINAL FLOWER1*; *RA*, *RAMOSA*; *VRS*, *Six-rowed spike*; *SRA3*, *SISTER OF RAMOSA3*; *TPP4*, *Trehalose-P-phosphatase 4*; *FZP*, *FRIZZY PANICLE*; *BD1*, *BRANCHED SILKLESS1*; *COM*, *COMPOSITUM*; *TtBH1*, *BRANCHED HEAD1*; *WFZP*, *wheat FZP*; *TB*, *TEOSINTE BRANCHED*; *REP1*, *RETARDED PALEA1*; *BAD1*, *BRANCH ANGLE DEFECTIVE 1*; *BDI1*, *BRANCHED AND INDETERMINATE SPIKELET 1*; *APO*, *ABERRANT PANICLE ORGANIZATION*; *TAW1*, *TAWAWA1*; *SPL*, *Squamosa Promoter Binding Like Protein*; *SNB*, *SUPERNUMERARY BRACT*; *IDS1*, *INDETERMINATE SPIKELET 1*; *SID1*, *Sister of INDETERMINATE SPIKELET 1*; *INT-M/DUB1*, *INTERMEDIUM-M/DOUBLE SEED1*; *AP2L2*, *APETALA 2-Like gene 2*; *Q*, *APETALA 2-Like gene 5*.

## Data Availability

Not applicable.

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
