# Peer review of "Molecular Insights into Inflorescence Meristem Specification for Yield Potential in Cereal Crops"

_ijms, 2021, doi:10.3390/ijms22073508_

Round 1
Reviewer 1 Report
The introduction part is to short, is must be presented in a more extensive way. Also, the authors must underlines how their research contributes to the knowledge in the field.
The key regulators from table 1 and table 2 are only in abbreviated form. It must be put on tables legend and in an extensive form.
I do not agree that the bibliography to be put in the conclusion part. The conclusions must be only authors point of view related to their review subject.
Author Response
Response to the comments by Reviewer #1
Comment 1: The introduction part is to short, is must be presented in a more extensive way. Also, the authors must underlines how their research contributes to the knowledge in the field.
Response: Thanks for this suggestion. We have added more general statements of inflorescence development in Introduction and revised the effects of inflorescence meristem development in architecture diversity (Please see Lines 37-43). We have also highlighted most recent research progresses in Triticeae crops, which are often ignored in reviews of inflorescence architecture (Line 45-50). By summarizing developmental diversity of inflorescences in cereal crops at the molecular level, we proposed various possibilities to improve grain yield of cereal crops (Lines 75-84).
Comment 2: The key regulators from table 1 and table 2 are only in abbreviated form. It must be put on tables legend and in an extensive form.
Response: Thanks, we have added the abbreviations of all gene names that mentioned in Tables 1 and 2 (please see the legends of Tables in Lines 211-217; 420-426).
Comment 3: I do not agree that the bibliography to be put in the conclusion part. The conclusions must be only authors point of view related to their review subject.
Response: We agree that the conclusion part should not include the cited papers. In this manuscript, part 5 contains two paragraphs: the first one is the summary of this review, which did not cite any publications; the second paragraph is the further perspectives, as suggested, we removed the bibliography citation of QTLs (ref. [13, 14]). Other cited publications indicate the future crop breeding strategies [149], wheat/barley genome and pan-genome resources [150–153], and some key technologies, such as CRISPR/Cas9 and single-cell RNA sequencing [154,155], which is important for further molecular functional investigations of cereal inflorescence development. Thus, we kept these citations in ‘further perspectives’ part, not in ‘Conclusion’.
Reviewer 2 Report
As the title indicates that Molecular insights into inflorescence meristem specification for yield potential in cereal crops, the comparative study among Poaceae grain species is the value of the article. Review is focused well with extensive coverages of information from different groups. Illustration is displayed well visually. Generally, it is recommended to share ideas on IM. Citation style check is encouraged in minor points.
Author Response
Response to the comments from Reviewer #2
As the title indicates that Molecular insights into inflorescence meristem specification for yield potential in cereal crops, the comparative study among Poaceae grain species is the value of the article. Review is focused well with extensive coverages of information from different groups. Illustration is displayed well visually. Generally, it is recommended to share ideas on IM. Citation style check is encouraged in minor points.
Response: Thanks for the positive comments. We focus on the IM identity and development in cereals, which provides the key genetic/molecular insights into crop breeding and yield improvement. As suggested, we have revised the errors of each citation style, and corrected other minor mistakes throughout the manuscript.